# Arginase 2 Deficiency Promotes Neuroinflammation and Pain Behaviors Following Nerve Injury in Mice

**DOI:** 10.3390/jcm9020305

**Published:** 2020-01-22

**Authors:** Yuhua Yin, Thuỳ Linh Phạm, Juhee Shin, Nara Shin, Dong-Wook Kang, Sun Yeul Lee, Wonhyung Lee, Cuk-Seong Kim, Sang Ryong Kim, Jinpyo Hong, Dong-Woon Kim

**Affiliations:** 1Department of Medical Science, Chungnam National University School of Medicine, Daejeon 35015, Korea; yoonokhwa527@gmail.com (Y.Y.); ptlinh@hpmu.edu.vn (T.L.P.); nongddangol@gmail.com (J.S.); s0870714@gmail.com (N.S.); ehddy1313@naver.com (D.-W.K.); cskim@cnu.ac.kr (C.-S.K.); 2Department of Anatomy and Cell Biology, Brain Research Institute, Chungnam National University School of Medicine, Daejeon 35015, Korea; 3Department of Physiology, Chungnam National University School of Medicine, Daejeon 35015, Korea; 4Department of Anesthesia and Pain Medicine, Chungnam National University Hospital, Daejeon 35015, Korea; neoquack@gmail.com (S.Y.L.); whlee@cnu.ac.kr (W.L.); 5School of Life Sciences, BK21 Plus KNU Creative BioResearch Group, Institute of Life Science & Biotechnology, Brain Science and Engineering Institute, Kyungpook National University, Daegu 41566, Korea; srk75@knu.ac.kr

**Keywords:** macrophages, microglia, neuropathic pain, neuroinflammation, arginase 2

## Abstract

Microglia, the resident macrophages, act as the first and main form of active immune defense in the central nervous system. Arginase 2 (Arg2) is an enzyme involved in L-arginine metabolism and is expressed in macrophages and nervous tissue. In this study, we determined whether the absence of Arg2 plays a beneficial or detrimental role in the neuroinflammatory process. We then investigated whether the loss of Arg2 potentiated microglia activation and pain behaviors following nerve injury-induced neuropathic pain. A spinal nerve transection (SNT) experimental model was used to induce neuropathic pain in mice. As a result of the peripheral nerve injury, SNT induced microgliosis and astrogliosis in the spinal cord, and upregulated inflammatory signals in both wild-type (WT) and Arg2 knockout (KO) mice. Notably, inflammation increased significantly in the Arg2 KO group compared to the WT group. We also observed a more robust microgliosis and a lower mechanical threshold in the Arg2 KO group than those in the WT group. Furthermore, our data revealed a stronger upregulation of M1 pro-inflammatory cytokines, such as interleukin (IL)-1β, and a stronger downregulation of M2 anti-inflammatory cytokines, including IL4 and IL-10, in Arg2 KO mice. Additionally, stronger formation of enzyme-inducible nitric oxide synthase, oxidative stress, and decreased expression of CD206 were detected in the Arg2 KO group compared to the WT group. These results suggest that Arg2 deficiency contributes to inflammatory response. The reduction or the loss of Arg2 results in the stronger neuroinflammation in the spinal dorsal horn, followed by more severe pain behaviors arising from nerve injury-induced neuropathic pain.

## 1. Introduction

Macrophages play key roles in the pathogenesis and regulation of the inflammatory reaction [1]. Activated macrophages trigger the immune system and drive immune responses in different pathways depending on the inflammatory condition [2]. When macrophages are prone to release pro-inflammatory cytokines, they contribute to the development of inflammation and are classified as M1 macrophages. They are considered M2 macrophages when they produce anti-inflammatory mediators and potentiate cell proliferation, tissue repair, and the healing process [3]. These distinct aspects of macrophage biology are fundamentally driven by the metabolism of L- arginine via two different pathways [4]. In M1 macrophages, L-arginine is metabolized to nitric oxide (NO) and citrulline under the action of nitric oxide synthase (NOS) expressed by the M1 phenotype. This pathway prevents cell proliferation, causes cellular toxicity, and facilitates inflammation. By contrast, M2 macrophages with strong expression of arginase, which hydrolyzes L-arginine to ornithine and urea, contribute to cell proliferation and resolution of inflammation. Both NO and ornithine are generated from the same L-arginine cellular substrate by NOS and arginase, respectively. Hence, these two opposite biosynthetic pathways limit the available L-arginine cellular substrate of the other. It has been demonstrated in many studies of various diseases that arginase via arginine/ornithine metabolism reduces the production of NO via NOS, by limiting the availability of the L-arginine cellular substrate and subsequently impairing the inflammatory state [4,5].

Arginase 1 (Arg1) and Arginase 2 (Arg2) are two isoforms of arginase. Although both Arg1 and Arg2 participate in ornithine and urea synthesis from L-arginine, these two isoforms differ in gene encoding and location [6]. These features may lead to their different functions and contribution to the M1 and M2 macrophage phenotypes. Arg1 is expressed predominantly in the liver as a component of the urea cycle, whereas Arg2 is found mainly in the kidney, endocrine glands, and notably, in macrophages and the brain [7]. The role of Arg1 identified from early studies suggests that it supports the anti-cytotoxic function of hepatocytes. A deficiency of Arg1 results in a urea cycle disorder, liver dysfunction, and neurocognitive deficits [7,8,9,10]. These findings strongly implicate that Arg1 contributes to the M2 macrophage phenotype in the inflammatory process due to their protective role. Initially, Arg2 was believed to play a similar role as Arg1, because of their similar structure and function in L-arginine metabolism. However, some recent studies have revealed the detrimental role of Arg2 in several pathological disorders and diseases, particularly in metabolic disorders [11]. Arg2 has been demonstrated to contribute to diabetic renal injury in mice with diabetes [12]. A recent study published in 2019 showed the detrimental role of Arg2 overexpression in the pathogenesis of osteoarthritis [13] when it upregulated pro-inflammatory cytokines in osteoarthritis cartilage. However, those data also revealed that the overexpression of Arg2 significantly reduces interleukin (IL)-1β-caused NO production, which appears as a pro-inflammatory marker, and that the knockdown of Arg2 increases IL-1β-induced NO levels [13]. Taken together, these studies suggest that Arg1 and Arg2 may play similar or different roles depending on the cells or organs in which they are located [11].

Despite these suggestions, the role and mechanism of action of Arg2 are still poorly understood, particularly in nervous system diseases, and the neurological pathology remains unclear. In this study, we investigated the contribution of Arg2 to neuropathic pain induced from peripheral nerve injury. We hypothesized that Arg2 attenuates neuroinflammation, and the lack of Arg2 potentiates the development of inflammation, due to its effect on limiting NO production. Many recent studies have demonstrated that neuropathic pain results from chronic neuroinflammation [14,15]. Hence, we attempted to clarify whether a lack of Arg2 could regulate neuropathic pain and change pain behaviors. It is also important to note that microglia, the macrophage-like cells in the central nervous system (CNS), have been demonstrated in numerous studies to play a major role in either induction or resolution of pain according to their M1 and M2 microglial phenotypes [16,17]. The M1 and M2 microglial phenotypes contribute to the neuroinflammatory condition by releasing pro- and anti-inflammatory cytokines respectively, and result in pain behaviors. Notably, Arg2 is expressed in macrophage cells and nervous tissue [7]. The role of Arg2 in L-arginine metabolism and the evidence that a lack of Arg2 potentiates neuroinflammation may suggest its crucial role in the inflammatory process in the spinal cord following nerve injury. In this study, we used the spinal nerve transection (SNT) model to induce neuropathic pain in wild-type (WT) and Arg2 knockout (KO) mice. We assessed and compared the pain behaviors of the mice, the inflammatory condition in the spinal cord, and the expression of inflammatory markers between WT and Arg2 KO mice. The data obtained in this study demonstrate the beneficial role of Arg2 in the pathogenesis of neuropathic pain.

## 2. Materials and Methods

### 2.1. Experimental Animals

Male C57BL/6 mice (21–23 g) used in this study were obtained from Daehan Bio Link (DBL, Chung-buk, Korea). C57BL/6 mice with knockdown of the Arg2 gene were kindly provided by Dr. William O’Brien, with confirmation of the Arg2 silenced gene. Genotyping was performed by polymerase chain reaction as previously described. The Arg2 KO and C57BL/6 WT mice were housed in divided cages under a controlled 12:12 h light:dark cycle at 23 °C. The mice were kept for at least 7 days under these conditions before surgery. All experiments were carried out with the approval of the Animal Care and Use Committee of Chungnam National University (CNU-017-A0046) and were in accordance with the ethical guidelines of the National Institutes of Health and the International Association for the Study of Pain.

### 2.2. Pain Model

All mice were 8 weeks of age (21–23 g) at the time of surgery, and efforts were made to limit distress to the animals. The mice were anesthetized with an intraperitoneal injection (i.p.) of Avertin (2,2,2-tribromoethanol, 50% w/v in tertiary amyl alcohol, diluted 1:40 in distilled water; 20 mL/kg, i.p.; Sigma-Aldrich, St. Louis, MO, USA). Spinal nerves were transected as described previously [18], with the following minor modification. Following surgical preparation, the left paraspinal muscles were separated by an incision from the spinal processes at L4 to the S2 levels. The L6 transverse process was partially removed, and the L4 and L5 spinal nerves were identified. Then, the L5 spinal nerve was separated and transected. After surgery, the mice recovered on a heating pad at 27 °C. A similar surgery procedure excepting nerve transection was conducted to the mice with WT and Arg2 KO genotype as the sham groups.

### 2.3. Pain Behavioral Test

Mechanical allodynia was assessed by measuring foot withdrawal thresholds in response to mechanical stimuli to the hind paw. The withdrawal threshold was determined using the up-down method with a set of von Frey filaments. In detail, the mice were placed on a metal mesh floor covered with clear plastic cages (18 × 8 × 8 cm) and allowed a 20 min period for habituation. Mechanical stimuli were applied with nine different von Frey filaments ranging from 0.008–1.4g (0.008, 0.02, 0.04, 0.07, 0.16, 0.4, 0.6, 1, and 1.4 g). Stimuli were applied with a von Frey filament in 3–4 s trials, each of which was repeated four times on each hind paw at approximately 5 min intervals. The 0.6 g filament stimulus was applied first. If a positive response occurred, the next smaller von Frey filament was used; if a negative response occurred, the next larger filament was applied.

### 2.4. CatWalk Automated Gait Analysis in SNT Mice

The CatWalk automated gait analysis was used to evaluate gait parameters in the SNT-induced neuropathic pain model. The CatWalk analysis system consists of a glass walkway, which contains light from a white fluorescent source. The light rays from this source exhibit complete internal reflection. When an object touches the glass runway, the light is reflected downwards, where it is detected by a video camera, as the mice travel down a glass-floored walkway located in a darkened tunnel. In general, mice cross the CatWalk runway easily and at a constant speed. The downward pressure of each paw illuminates the fluorescent light from the glass platform, and the video camera detects and records the light signals. Then, CatWalk XT software automatically analyzes various gait parameters, such as paw print area, duty cycle, and the stance phase. In this study, we examined two parameters (print area and single stance, Appendix A) to study the role of Arg2 in SNT mice. The print area was measured by calculating the surface area contacting the glass floor and the single stance was part of the hind paw step cycle where the contralateral hind paw does not touch the glass plate or where the ipsilateral hind paw touches the glass plate. This signal is then digitized, which allows analysis by the CatWalk program software (CatWalk XT version 10.5.505, Noldus Information Technology, Wageningen, The Netherlands).)

### 2.5. Western Blot Analysis

Mice were sacrificed on day 7 after the SNT surgery with sodium pentobarbital (50 mg/kg, i.p.). Shortly afterwards, the vertebral column was exposed and was then removed from the mice. The spinal cord tissue was extracted using hydraulic pressure applied to the caudal vertebral canal, whereupon the tissue was washed in phosphate-buffered saline (PBS) to remove any remaining blood contaminants. The enlarged segments L4–L5 (0.5 cm) were carefully dissected out. The halves of lumbar segments ipsilateral to the SNT or sham surgery were collected. The dorsal part of the spinal cord was separated from the ventral part. Afterwards, the tissues were quickly frozen in liquid nitrogen and then stored at −20 °C until analyses.

In brief, the tissue lysed from each sample was resolved by 10% or 15% sodium dodecyl sulfate-polyacrylamide gel electrophoresis. The proteins were transferred to nitrocellulose membranes and blocked with 5X Phospho-BLOCK Solution (Translab, Daejeon, Korea, TLP-115.1P) diluted to 1X in distilled water. The membranes were probed overnight with primary antibodies (Arg2, 1:50, sc-393496; Santa Cruz, Dallas, TX, USA; CD206, 1:100, 18704-1-AP; Proteintech, Rosemont, IL, USA; inducible nitric oxide synthase iNOS, 1:100, BD610329BD; BD Biosciences, San Jose, CA, USA) at 4 °C. The immune complexes were identified using an enhanced chemiluminescence detection system (Habersham, Little Chalfont, UK).

### 2.6. Immunostaining Analysis

Immunohistochemistry was performed 7 days after surgery. The mice were anesthetized with sodium pentobarbital (50 mg/kg, i.p.) and perfused transcardially with heparinized phosphate-buffered saline (PBS, pH = 7.4), followed by perfusion with 4% paraformaldehyde for 15 min. The lumbar enlarged (L4–L6) regions of the spinal cord were removed, post-fixed in the same fixative, and cryoprotected with 30% sucrose in 0.05 M PBS. After 2 days, a cryostat was used to cut the L5 spinal cord segments into 30 μm transverse sections, which were preserved in storage buffer. After incubating the tissues with blocking buffer (5% normal serum/0.3% Triton X-100) for 1 h to prevent nonspecific binding, the sections were incubated with primary antibodies (Iba-1, 1:400, Wako Pure Chemical, Osaka, Japan, 019-19741, RRID:AB_839506; anti-NeuN, 1:100, Millipore, Milford, MA, USA; MAB377, RRID:AB_2298772; anti-GFAP, 1:1000, Millipore, AB5804, RRID:AB_2109645, CD206, 1:100, Proteintech, Rosemont, IL, USA, 18704-1-AP; iNOS, 1:100, BD Bioscience, Frankin Lakes, NJ, USA, BD610329) diluted in blocking buffer, as previously reported [19]. After washing, the sections were incubated with biotinylated secondary antibody for 2 h at room temperature with a mixture of fluorescein isothiocyanate-conjugated anti-mouse IgG and Cy3-conjugated anti-rabbit IgG antibodies (1:200, Jackson ImmunoResearch Labs, West Grove, PA, USA). The sections were mounted with Vectashield (Vector Laboratories, Burlingame, CA, USA) and fluorescent images were obtained with a confocal microscope. Three independent slices were taken and the area of laminae I and II were imaged in each slice. The immunodensities in the graphs were quantified by Image J software. The selected area (laminae I and II of the spinal dorsal horn) in each slice was cropped to the same size, and the immunodensities were then calculated following the ideal setting. The data are presented as the mean value of the immunoreactivity from three independent slices.

### 2.7. Quantitative Polymerase Chain Reaction (qPCR)

Total RNA was isolated from spinal dorsal horn tissues (L4–L5 segment, 0.5 cm) using TRIzol reagent (Invitrogen, Carlsbad, CA, USA) according to the manufacturer’s instructions. The concentration and purity of the RNA were assessed using a NanoDrop spectrophotometer (Thermo Fisher Scientific, Waltham, MA, USA). cDNA was synthesized in a 20 μL reaction with TOPscript RT DryMix (Enzynomics, Daejeon, Korea). Quantitative polymerase chain reaction (qPCR) analysis was performed under the following conditions: 95 °C for 10 min, 40 cycles of 95 °C for 15 s, and 60 °C for 1 min using the AriaMx Realtime PCR System (Agilent Technologies, Santa Clara, CA, USA). The primer sequences (Cosmogenetech, Daejeon, Korea) used for qPCR were as follows: mouse GAPDH, forward: 5’-ACC CAG AAG ACT GTG GAT GG -3’ and reverse: 5’-CAC ATT GGG GGT AGG AAC AC -3’, mouse IL-1β, forward: 5’-TTG TGG CTG TGG AGA AGC TGT -3’ and reverse: 5’-AAC GTC ACA CAC CAG CAG GTT -3’, mouse IL-4, forward: 5’-CCT CAC AGC AAC GAA GAA CA -3’ and reverse: 5’-ATC GAA AAG CCC GAA AGA GT -3’, mouse IL-10, forward: 5’-ACTTGCTATGCTGCCTGCTCT -3’ and reverse: 5’-ATGTTGTCCAGCTGGTCCTT -3’, and mouse iNOS, forward: 5’- GGC AAA CCC AAG GTC TAC GTT -3’, and reverse: 5’-TCG CTC AAG TTC AGC TTGGT -3’. The mRNA levels of each target gene were normalized to that of GAPDH mRNA. The fold-changes in the mRNA levels were calculated using the 2^−ΔΔ*C*t^ method, as previously described [20].

### 2.8. Reactive Oxygen Species (ROS) Detection Assay

Superoxide anion levels in the spinal cord were determined using dihydroethidium (DHE; Thermo Fisher Scientific), and Mitosox (M36008, Thermo Fisher Scientific) as previously described [21]. Spinal cord sections were incubated with DHE (1 µM) or Mitosox (1 µM) at room temperature for 5 min and mounted on slides. The sections were imaged using a confocal microscope (Leica Microsystems, Buffalo Grove, IL, USA).

### 2.9. Statistical Analysis

Results from the behavioral study were analyzed by two-way analysis of variance (ANOVA), followed by Tukey’s post hoc test. The results of immunoblot analysis and immunohistochemical analyses were statistically analyzed by one-way ANOVA. All data are presented as means ± standard error of the mean (SEM). Statistical analyses were performed using GraphPad Prism 6 (GraphPad Software Inc., La Jolla, CA, USA). In all analyses, significance was set at *p* < 0.05.

## 3. Results

### Pain Behaviors from SNT-Induced Neuropathic Pain in WT and Arg2 KO Mice

To assess whether SNT successfully induced neuropathic pain-like behaviors in the mice, we conducted von Frey filament tests and Catwalk analysis at different time points after surgery in the sham, WT, and Arg2 KO groups of mice. The von Frey filament tests were performed on the ipsilateral side (left hind paw) of mice beginning on day 3 post-surgery and repeated at 5, 7, 10, and 14 days after surgery. The mechanical thresholds of mice in the SNT WT group (*n* = 6) were effectively reduced compared to those in the sham group (*n* = 5). The mechanical thresholds present the sensitivity of the mice to an innocuous mechanical stimuli. The lower the mechanical threshold observed in the mice, the more sensitive to a mechanical stimuli they are. This mechanical hypersensitivity is one of the most common symptoms in neuropathic pain patients. A reduction in the mechanical threshold was recorded in the SNT WT group early on day 3 post-surgery and was maintained up to 14 days after surgery (Figure 1). Interestingly, the similar von Frey tests conducted in parallel to the SNT Arg2 KO group showed significantly lower mechanical thresholds, compared to the SNT WT group. All mechanical threshold values at 3, 5, 7, 10, and 14 days post-surgery decreased in SNT Arg2 KO mice compared to those in WT mice. Therefore, we initially confirmed that both groups of mice were more sensitive to an innocuous stimuli. This means that the SNT procedure successfully induced mechanical hypersensitivity in the mice. It is worth noting that mechanical hypersensitivity was more serious in Arg2 KO mice than in WT mice.

For additional evidence regarding pain behaviors, we also assessed gait variations as another sign of pain in mice. CatWalk track analysis has been shown to be an effective method to evaluate locomotor dysfunction affected by neuropathic pain-like behaviors after nerve injury [22]. The results obtained from CatWalk analysis were correlated with the von Frey tests, indicating the more serious pain-like behaviors in the SNT Arg2 KO mice. Figure 2A shows representative images of the footprint area of the right and left hind paws (ipsilateral and contralateral sides), illustrating the complete surface area of the hind paw once it touched the glass plate. No significant difference in print area was detected between the ipsilateral and contralateral high paw in the sham group. However, the print area of the ipsilateral hind paw was much more reduced in length and width in SNT WT mice, compared with the contralateral side, which appeared to be larger than normal in the sham group. We did not observe the ipsilateral hind paw print area of the Arg2 KO group mice because they did not place their left paw on the glass plate. By contrast, the print area of the right paw was even larger than that in the WT group. Quantitative data of the footprint area presenting the proportion of ipsi/contralateral hind paw print area is shown in Figure 2B. These data suggest that the proportion in Arg2 KO mice was the lowest 10%, while it was approximately 50% in WT mice, and 80% in the sham group. These results are likely due to the pain in the ipsilateral hind paw. Thus, the mice placed almost all of their body pressure on the contralateral paw to reduce pain. The second parameter we measured in the CatWalk analysis was the single stance assessed by the duration of a single hind paw touching the glass plate (Figure 2C,D). In the sham group (*n* = 3), the proportion of ipsi/contralateral was almost 100%. However, in both the Arg2 KO (*n* = 3) and WT (*n* = 4) groups, this proportion decreased, and the greatest decrease occurred in the Arg2 KO mice (approximately 25% in SNT Arg2 KO mice and 75% in SNT WT mice), as shown by the image analysis (Figure 2C) and quantitative data (Figure 2D). The additional images of the footprint area and single stance obtained from other mice in each group are shown in Appendix A, respectively. Taken together, we concluded that a neuropathic pain model, in which the mice presented pain-like behaviors, was successfully induced by the SNT procedure in both groups of mice. Notably, the level of mechanical hypersensitivity was more serious and the locomotor dysfunction behaviors were more pronounced in the SNT Arg2 KO mice than in the WT mice.

Following confirmation of neuropathic pain behavior in the mice, we examined the expression of Arg2 in both groups. We checked the Arg2 mRNA levels in the spinal dorsal horns of WT mice at different time points after surgery by real-time (RT)-PCR. Figure 3A shows that Arg2 mRNA increased the most on days 3 and 7 after surgery and this increased level was not significant on day 14. We also compared the protein expression of Arg2 in the spinal dorsal horns of mice on day 7 post-surgery among the sham WT, the sham Arg2 KO, the SNT WT, and the SNT Arg2 KO mice by western blotting (Figure 3B). The blots and corresponding quantification of band intensity indicate stronger expression of Arg2 in the SNT WT mice than in the sham WT mice, which was consistent with the qPCR data shown in Figure 3A. Additionally, the results also show that the protein expression of Arg2 in both the sham and the SNT Arg2 KO mice was very low. This serves as confirmation of KO of Arg2.

Since Arg1 and Arg2 are two isoforms of the Arginase enzyme system, sharing the similar function in L-arginine metabolism, questions arose as to whether KO of Arg2 affected the basal expression level of Arg1 under normal conditions and whether this effect played any roles in modulating inflammatory response in the spinal cord of the SNT Arg2 KO mice. To address these questions, we attempted to assess the basal levels of Arg1 in both of the sham WT and the sham Arg2 KO mice and compared the changes in Arg1 in the injury groups after the SNT procedure. Figure 3C shows the gene expression of Arg1 in the spinal dorsal horns of the mice from the four groups: the (sham WT, SNT WT, sham Arg2 KO, and the SNT Arg2 KO), assessed by RT PCR. In the sham Arg2 KO, the basal level of Arg1 remained unchanged, compared to that in the sham WT. Alternatively, the deletion of Arg2 in Arg2 KO mice did not affect the basal mRNA level of Arg1 under normal conditions. However, Arg1 was downregulated in the injury groups, compared with the corresponding sham groups. Stronger downregulation of Arg1 was observed in the Arg2 KO mice. Corresponding well with the gene expression, the protein levels of Arg1 in the four groups measured by western blot (Figure 3D) showed the similar basal levels in the sham groups and increased reduction of Arg1 in the SNT Arg2 KO mice. These results indicate that the lack of Arg2 directly resulted in the stronger downregulation of Arg1 in the SNT Arg2 KO mice. This regulation seems to be consistent with the stronger downregulation of other anti-inflammatory markers in the SNT Arg2 KO mice, which will be presented and discussed later in the following sections.

Many studies have demonstrated the modulatory role of microglia and astrocytes in the pathogenesis of neuropathic pain [16,17]. Microgliosis and astrogliosis, which consist of cell proliferation, morphological changes, and activation, are the most common characteristics in the spinal cord following peripheral nerve injury [23]. As a result, the SNT procedure induced microgliosis and astrogliosis in the spinal cords of both groups of mice. We first examined and compared the levels of microgliosis and astrogliosis among the groups by immunohistochemical staining. Figure 4A,B are representative images of microgliosis and astrogliosis respectively, in the ipsilateral spinal dorsal horns of mice from the sham, SNT WT, and SNT Arg2 KO groups. Microglia and astrocytes are commonly activated the most in laminae I and II (the regions shown in the white dotted frames of the upper panels in Figure 4A,B) of the ipsilateral spinal dorsal horn segments L4–L5 following nerve injury. Immunostaining was performed on spinal cord sections on days 3, 7, and 14 post-surgery. No significant microgliosis or astrogliosis was observed in the sham group. In the SNT WT and Arg2 KO groups, we observed the activation of microglia and astrocytes in the spinal cord at every time point. The strongest microgliosis and astrogliosis were recognized on day 7 post-surgery in both groups. Notably, the most robust activation of these cells occurred in the spinal cord of Arg2 KO mice. We also showed the quantification of Iba-1 (microglial marker) and GFAP (glial fibrillary acidic protein, astrocytic marker) immunoreactivities, in comparison with the WT and Arg2 KO at corresponding times (Figure 4C,D), which show larger values for the Arg2 KO group compared to the WT group at every time point and confirmed that the largest values were reached on day 7 post-surgery in each group.

Based on the intensity of the microgliosis and astrogliosis described above, we subsequently checked and compared the expression of inflammatory signals between SNT WT and Arg2 KO mice. As mentioned above, NO metabolized from L-arginine via NOS is toxic to cells and facilitates the development of inflammation [6,9,24]. Of note, NOSs are a family of enzymes, including several isoforms that differ in location and organ function [25]. iNOS is an isoform located predominantly in macrophages that plays a role in the immune system and functions in inflammatory pathogenesis [24,26]. iNOS is also a pro-inflammatory marker. In contrast, CD206, a mannose receptor expressed on the surface of macrophages, is considered as an anti-inflammatory marker that promotes the resolution of inflammation by controlling the levels of molecules released during the immune response that can be damaging to host tissues [27]. Thus, it is necessary to check and compare the levels of iNOS/CD206 to examine the regulation of pro-/anti-inflammatory markers induced following SNT surgery in the spinal cords of the mice. Figure 5 shows the immunohistochemical staining results of the spinal cord sections on day 7 post-surgery. The representative images also show a number of iNOS and CD206 expressing cells, which were overlapped with microglia (shown by the Iba-1 microglial marker), demonstrating the involvement of microglia in the inflammatory reaction in the spinal cord of the mice. Figure 5A shows stronger expression of iNOS in the ipsilateral spinal dorsal horns of SNT WT and Arg2 KO mice, compared to those of sham mice. Notably, the most intense expression of iNOS was found in Arg2 KO mice. These data suggest upregulation of iNOS caused by the nerve injury. In contrast to iNOS, expression of CD206 was downregulated in SNT mice (Figure 5B). In addition, the largest decrease in CD206 expression occurred in Arg2 KO mice.

The quantification of iNOS and CD206 expression is shown in Figure 5C–F and is correlated with the immunohistochemical staining images. Figure 5C,D show the upregulation of iNOS immunoreactivity in the spinal dorsal horn, and iNOS^+^/Iba-1^+^ co-expressing cells/total Iba-1 expressing cells, respectively. Figure 5E,F illustrate CD206 quantification. Both total CD206 immunoreactivity in the spinal cord and CD206 immunoreactivity in Iba-1-positive cells were downregulated in SNT WT and Arg2 KO mice. For more convincing evidence, we also quantified iNOS and CD206 protein expression in the spinal dorsal horns by western blotting (Figure 5G,H). Consistent with the immunostaining, the western blot results showed an increased iNOS expression from the sham to the Arg2 KO group, whereas CD206 decreased (Figure 5G). In addition, quantification of band intensity of iNOS and CD206 in Figure 5H also served as proof of these modulations. Collectively, iNOS and CD206 were upregulated and downregulated respectively, in SNT mice following surgery. Notably, the greatest number of regulatory changes was found in Arg2 KO mice.

As mentioned above, NO, a metabolic product of L-arginine via iNOS and a subset of ROS, is toxic to cells, and plays a harmful role in inflammatory pathogenesis [28]. ROS describes a number of reactive molecules and free radicals derived from molecular oxygen that are generated during mitochondrial oxidative metabolism [29]. When excess ROS is produced in cells, there is an imbalance between oxidants and antioxidants, resulting in cellular damage, apoptosis, and neuroinflammation, which causes excessive inflammation and neurodegeneration [30]. Thus, we further investigated the level of ROS production in ipsilateral spinal dorsal horns of the three groups of mice. The spinal cord sections were stained with Mitosox (mitochondrial ROS marker) and DHE (cytosolic ROS marker). The results are shown in Figure 6. As expected, the images of spinal cords stained with Mitosox or DHE (Figure 6A,B) show increased production of ROS in both the mitochondria and the cytoplasm of cells in the spinal cords of the SNT mice, compared with the sham mice (control group). The ROS production in the three groups are quantified and compared in Figure 6C,D. Once again, our data revealed that the enhancement of the superoxide formation (ROS production) was strongest in the SNT Arg2 KO mice.

It has been well documented that diverse causes of neuropathic pain are associated with excessive inflammation in both the peripheral and central nervous system that may contribute to the initiation and the maintenance of persistent pain [15]. The balance between pro- and anti-inflammatory markers indicates the predominance of M1 or M2 macrophages during the inflammatory process. Inflammatory markers in the spinal cord were assessed via qRT-PCR. We quantified the gene expression of pro-inflammatory markers IL-1β and iNOS, as well as the anti-inflammatory markers IL-4 and IL-10. The qPCR results for these markers (Figure 7A,B) were consistent with our assessments regarding the regulation of inflammatory markers in SNT mice. IL-1β and iNOS levels were upregulated in SNT WT and Arg2 KO mice and the highest levels were observed in the Arg2 KO group. In contrast to IL-1β and iNOS, IL-4 and IL-10, which are produced by M2 macrophages, were downregulated following surgery, especially in Arg2 KO mice (Figure 7C,D). The mRNA levels of these inflammatory markers in the spinal dorsal horn were also assessed at the time point of 14 days post-surgery (Figure 7E–H). These levels remained not significantly different from the levels of day 7 tissues. This means the lack of Arg2 in the KO mice affected both the levels of pro- and anti-inflammatory mediators released in the spinal cord. These data serve as direct evidence for stronger upregulation and downregulation of pro- and anti-inflammatory markers respectively, in the spinal cord of the Arg2 KO mice, after the SNT procedure.

## 4. Discussion

This study investigated the effect of Arg2 deficiency on the inflammatory condition in the spinal cord and pain state of mice with a SNT neuropathic pain model. Microglia have been demonstrated to play key roles in the induction of neuropathic pain, which results from chronic neuroinflammation following peripheral nerve injury [16,17,23]. Nerve injury leads to the release of the first microglial activators from damaged sensory neurons, triggering activation of microglia, resulting in neuroinflammation in the spinal cord [17]. In this study, we applied the SNT experimental model, which has been widely and effectively used to induce neuropathic pain in mice. After the surgical procedure, robust microgliosis and astrogliosis were observed in the spinal cords of SNT WT and Arg2 KO mice. Microglia and astrocytes began to be activated on day 3 post-surgery and reached their maximum levels on day 7 post-surgery. These data correlated with numerous previous studies reporting that microgliosis normally reaches the maximum level on day 7 after activation [17]. That was why we used the samples of spinal cords on day 7 post-surgery for almost all analyses in this study. Interestingly, the levels of microgliosis and astrogliosis were higher in Arg2 KO mice than in WT mice. Consistent with this finding, the pain-like behaviors of Arg2 KO mice were more obvious than in WT mice, exhibited by the lower mechanical threshold and the CatWalk track analysis.

We questioned whether the lack of Arg2 in Arg2 KO mice could result in a more severe state of neuroinflammation as well as neuropathic pain in these mice. To address this question, we investigated the regulation of Arg2 in the spinal cords of mice on day 7 after SNT surgery. Surprisingly, Arg2 was upregulated after surgery in the WT mice, compared to the sham mice. One important point to address is to determine which one of the two isoforms, Arg1 or Arg2, played a role in effectively modulating inflammatory and pain state in the mice. We conducted experiments to verify the basal levels of Arg1 in both the sham WT and the sham Arg2 KO mice, and to verify the regulation of Arg1 in the injury groups following the SNT procedure. As indicated in Figure 3C, the basal level of Arg1 in the sham of the Arg2 KO remained unchanged compared to the sham WT, indicating that the deletion of Arg2 in the KO mice did not affect the level of Arg1 under normal conditions. However, after the SNT procedure, Arg1 was downregulated in the spinal dorsal horn of the injury mice, compared with the sham. Of note, Arg1 had a more increased downregulation in the SNT Arg2 KO than in the SNT WT, compared with the corresponding sham groups (sham WT and sham Arg2 KO). This indicates KO of Arg2 directly affected the abundance mRNA of Arg1 following surgery. Alternatively, Arg2 deficiency resulted in greater reduction of Arg1 induced by inflammation in the spinal cord, suggesting a modulatory role of Arg2.

In fact, several recent studies on various diseases and disorders have reported that upregulation of Arg2 has detrimental effects on the pathogenic process. For example, Arg2 overexpression has been demonstrated to contribute to the destruction of osteoarthritis cartilage and osteoarthritis pathogenesis [13]. In particular, in diseases associated with metabolic disorders, such as diabetes, Arg2 has been shown to play a causal role in the induction of diabetic renal injury [12]. In this study, we considered the upregulation of Arg2 in the spinal cord of SNT mice to be more detrimental, but then how could we explain the more severe pain state in mice with a lack of Arg2? A study on osteoarthritis (OA) demonstrated the detrimental role of Arg2 in promoting inflammation when it was overexpressed [13]. In that study, they have demonstrated that Arg2 overexpression in mouse joint tissues causes OA with synovitis and loss of glycosaminoglycans in articular cartilage followed by upregulation of pro-inflammatory products, although, their data show ablation of Arg2 promoted IL-1β caused NO production. However, this enhancement of NO production in Arg2*^−/−^*chondrocytes is not associated with the ability of Arg2 to regulate OA pathogenesis. As such, in the pathogenesis of OA, they have suggested a detrimental role for Arg2. In contrast to our concept, evidence that a lack of Arg2 could promote NO production may suggest a beneficial role of Arg2 in the pathogenesis of neuropathic pain since NO production is associated with the neuroinflammatory state. This suggests that Arg2 may play different roles in various pathological diseases, depending on the tissue in which it is located. Despite the detrimental role of Arg2 demonstrated in previous studies, there are no studies on the role of Arg2 in neurological diseases or neuropathic pain. It is reasonable for us to hypothesize that Arg2 may play a protective role in neuroinflammatory process in the CNS due to the direct evidence obtained in this study. In addition, upregulation of Arg2 in the spinal cords of SNT WT mice following surgery should be considered a signal of the anti-inflammatory marker for the following reasons.

As mentioned above, macrophages are usually categorized into pro-inflammatory M1 macrophages (classically activated) and anti-inflammatory macrophages (alternatively activated). This classification is based on in vitro studies, in which cultured macrophages were treated with molecules that stimulate their phenotypes [3]. Upon activation, microglia are capable of acquiring diverse phenotypes and, furthermore, are capable of shifting between the different phenotypes during an inflammatory response. M1 microglia are typically the initial responders to an insult. M1 microglia will produce proinflammatory cytokines, chemokines, and redox signaling molecules. Over time, the inflammatory response is shifted to be more anti-inflammatory, which is facilitated by M2 microglia. M2 microglia secrete anti-inflammatory cytokines and growth factors that promote attenuation of the inflammatory response and repair of damaged tissue. There is compelling evidence that crucial dynamics between M1 and M2 polarization during injury state and microglia have the capacity for multiple activation states. It suggests that there exists a spectrum that spans several different activation types with different functions [31].

In fact, in in vivo experiments, we could observe the increase of both the M1 and M2 phenotypes as evidence of microglial activation after surgery. However, the ratio of M1/M2 markers determines the state of inflammation. If this ratio increases, inflammation develops. Our data indicate a more severe inflammatory state in the mice lacking Arg2 than in WT mice, suggesting that the lack of Arg2 may potentiate neuroinflammation in the spinal cord. In contrast, the upregulation of Arg2 following surgery, which should be considered a signal of M2 marker, may promote an increased repair and healing process after inflammation in the SNT WT mice compared to in the SNT Arg2 KO mice.

We subsequently clarified our hypothesis by examining and comparing the state of neuroinflammation between SNT WT and Arg2 KO mice. The results obtained were all consistent. The lack of Arg2 in Arg2 KO mice caused upregulation of pro-inflammatory markers and downregulation of anti-inflammatory markers, compared to WT mice. iNOS, a pro-inflammatory marker, was expressed more robustly in Arg2 KO mice than in WT mice. iNOS and arginase share the same cellular substrate of L-arginine [4]. In Arg2 KO mice, the lack of Arg2 enhanced the availability of arginine for iNOS activity, potentiating iNOS expression and NO production. IL-1β, a powerful modulator of inflammation, was also more strongly upregulated in Arg2 KO mice. IL-1β is a pro-inflammatory cytokine involved in multiple pathways of inflammation [32]. In addition to iNOS and IL-1β, we also assessed the level of ROS produced in the two groups of mice. ROS is a component of the killing response of immune cells [28]. These molecules, produced as byproducts of oxidation reactions, have the potential to cause deleterious events in host cells. Interestingly, NO is a subset of ROS. The over-production of ROS in the spinal cords of mice following surgery was a result of excess neuroinflammation. Of note, in SNT Arg2 KO mice, ROS was more abundant than in the other groups. Collectively, our data demonstrated that the lack of Arg2 enhanced the levels of pro-inflammatory markers, such as iNOS, IL-1β, and ROS.

In contrast to the pro-inflammatory markers, stronger downregulation of anti-inflammatory markers was detected in Arg2 KO mice than in WT mice. CD206, a mannose receptor, plays a modulatory role in controlling the levels of molecules, such as glycoproteins, released into circulation during the inflammatory response, which can be damaging to host tissues [27]. Therefore, it is important to regulate their levels to control the state of inflammation. In our study, however, the level of CD206 was downregulated in SNT mice, and more significantly in Arg2 KO mice. Like CD206, the levels of IL-4 and IL-10, the other anti-inflammatory markers, also decreased the most in Arg2 KO mice. IL-4 and IL-10 have been shown to promote the production of either polyamines to induce proliferation of cells or proline to induce collagen production [33]. These activities are associated with the healing and the repair of tissues after inflammation. Notably, the lack of Arg2 caused the downregulation of these anti-inflammatory markers in the spinal cords of Arg2 KO mice after the SNT surgery. It is important to show that these regulations remained constant up to 14 days following surgery. Alternatively, the pro- and anti-inflammatory markers were more strongly up- and down-regulated respectively, in the SNT Arg2 KO mice, demonstrating the effect of Arg2 deficiency on both the inflammatory response and healing process in the spinal cord.

Our findings in this study serve as direct evidence for the protective role of Arg2 in the pathogenesis of neuroinflammation, although some previous studies have reported that Arg2 plays detrimental roles in the pathogenesis of other diseases. The distinct role of Arg2 may be due to differences in the contribution of Arg2 to host tissues where it is located. In this study, we investigated and demonstrated the role of Arg2 deficiency, which resulted in the more severe pain-like behaviors in Arg2 KO mice. Our data suggest a beneficial role of Arg2 when its absence enhanced the pain-like behaviors in mice and promoted the development of neuroinflammation. Unlike Arg1, which is an M2 macrophage marker, Arg2 is only beginning to be understood. Various studies on Arg2 function have explored the roles of Arg2 in different diseases, including the present study. Further studies are required to investigate and clarify the function and contribution of Arg2 in various disorders and diseases.

## Figures and Tables

**Figure 1 jcm-09-00305-f001:**
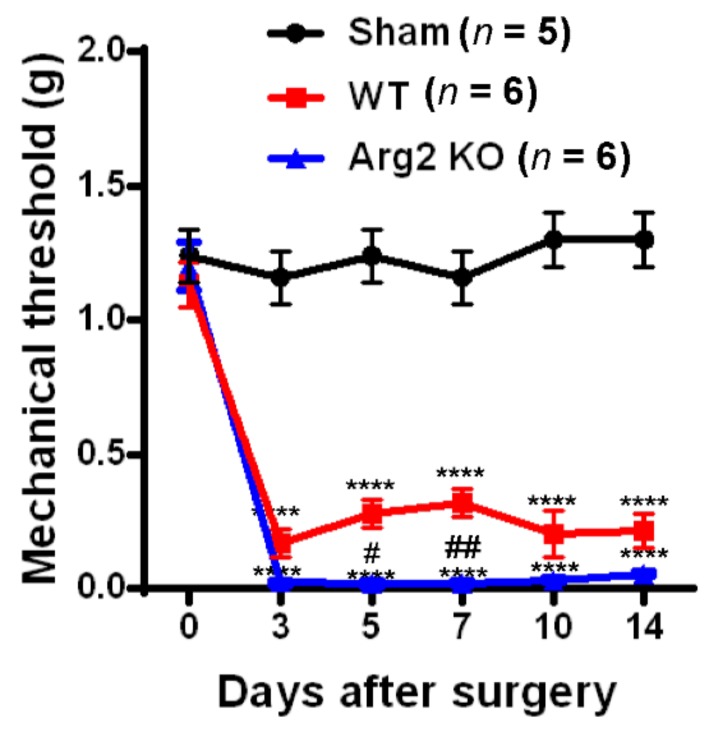
Pain behavioral tests of mice after surgery: The von Frey filament test. Von Frey filament tests were performed repeatedly on days 3, 5, 7, 10, and 14 post-surgery in the sham (*n* = 5), wild-type (WT; *n* = 6), and arginase 2 (Arg2) knockout (KO; *n* = 6) groups of mice. Following spinal nerve transection (SNT) surgery, the pain thresholds of the WT group decreased compared to the sham group. The lowest pain threshold was observed in the Arg2 KO group. Data are presented as means ± standard error of the mean (SEM) (two-way analysis of variance (ANOVA) with Tukey’s post hoc test, F_(10,82)_ = 13.30, **** *p* < 0.0001 versus Sham; (*n* = 6), ^##^
*p* < 0.01, and ^#^
*p* < 0.05 versus WT (*n* = 6).

**Figure 2 jcm-09-00305-f002:**
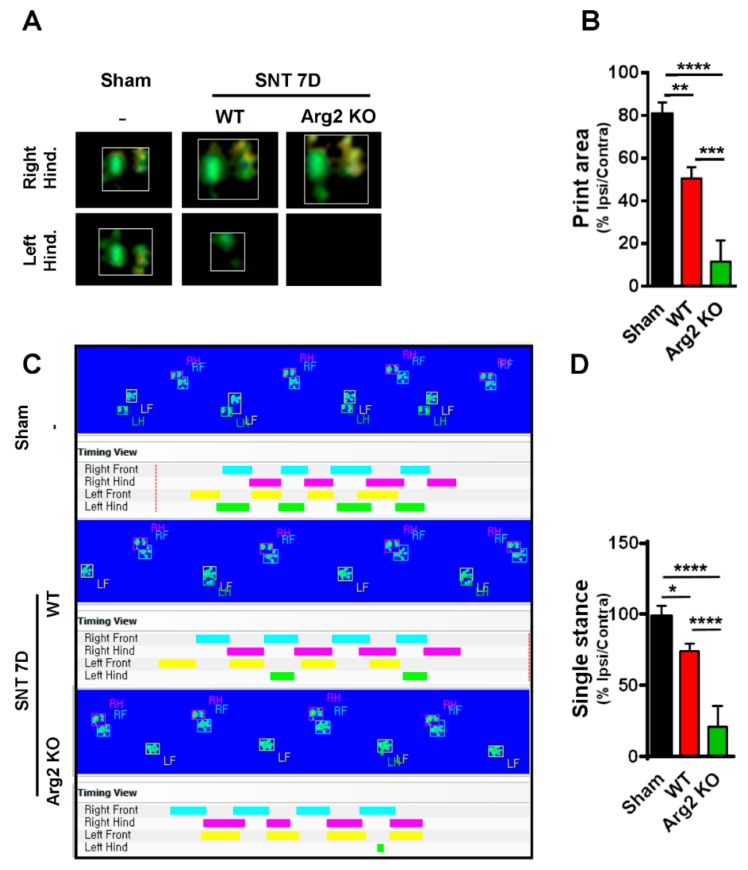
Representative images obtained in the CatWalk analysis on day 7 post-surgery. (**A**) Images of the footprint area of both left and right hind paws of mice. (**B**) Quantitative data for the footprint area compared among the three groups. (**C**) The single stance parameter measured by the duration time of a single hind paw touching the glass plate (ratio of length of green/pink bar). (**D**) Quantitative data for single stance parameter analysis compared among the three groups. (**B**, **D**) Sham (*n* = 3), WT (*n* = 4), and Arg2 KO (*n* = 3). Data are presented as means ± SEM (one-way ANOVA with Tukey’s post hoc test, F_(2,13)_ = 56.23, F_(2, 18)_ = 31.56, **** *p* < 0.0001, *** *p* < 0.001, ** *p* < 0.01, and * *p* < 0.05 versus Sham).

**Figure 3 jcm-09-00305-f003:**
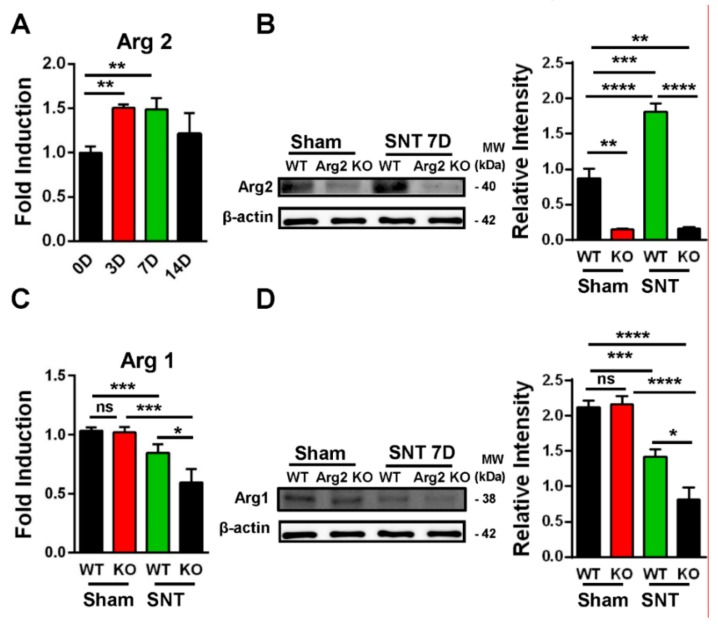
Upregulation of Arg2 in SNT WT mice and regulation of Arg1 after surgery. (**A**) The quantity of Arg2 mRNA increased in the spinal dorsal horn of WT mice on days 3, 7, and 14 post-surgery according to quantitative real-time polymerase chain reaction (qRT PCR) analysis. The maximum level was reached on day 3 and 7 post-surgery and was maintained until day 14. Data are presented as means ± SEM (one-way ANOVA with Tukey’s post hoc test, F_(3,19)_ = 6.751, ** *p* < 0.01 versus 0D). (**B**) Representative western blots and corresponding quantitative data for band intensity, comparing the expression of Arg2 in the spinal dorsal horn of mice on day 7 from the sham WT (*n* = 4), the sham Arg2 KO (*n* = 4), the SNT WT (*n* = 4), and the SNT Arg2 KO (*n* = 4) mice. Arg2 was expressed more strongly in the SNT WT mice, compared to the sham WT mice. (**C**) The gene expression levels of Arg1 of the sham WT, SNT WT, sham Arg2 KO, and the SNT Arg2 KO mice on day 7. Data are presented as means ± SEM (one-way ANOVA with Tukey’s post hoc test, F_(3,10)_ = 16.99, *** *p* < 0.001, * *p* < 0.05 versus WT or Sham). (**D**) The protein expression of Arg1 in the spinal dorsal horns of mice from the four groups (*n* = 4 per group) on day 7.

**Figure 4 jcm-09-00305-f004:**
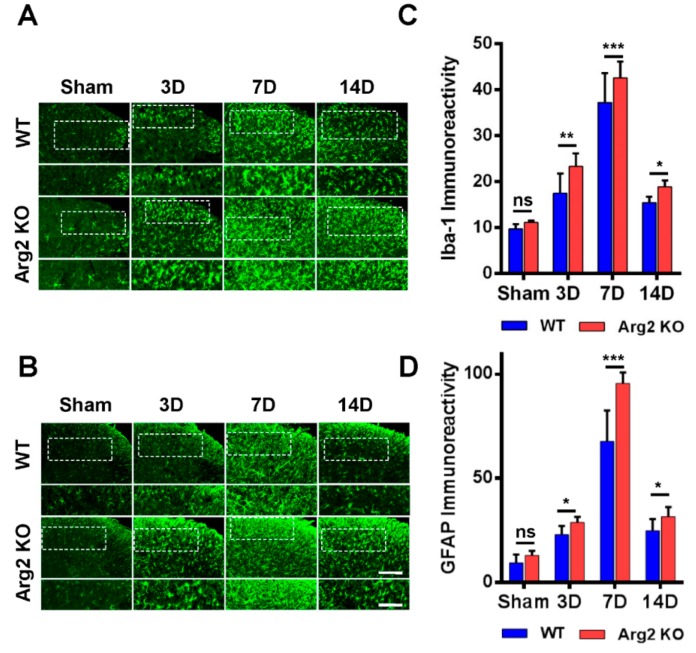
Robust microgliosis and astrogliosis were observed in the spinal cords of mice following SNT surgery. (**A**, **B**) Representative immunohistochemical staining images of spinal cord sections from mice on days 3, 7, and 14 post-surgery using Iba-1 (a microglial marker) and GFAP (glial fibrillary acidic protein, an astrocyte marker) antibodies. Microglia and astrocytes were activated in the spinal cords of SNT WT (*n* = 3) and Arg2 KO (*n* = 3) mice but were most robust in the Arg2 KO mice. The lower panel displays the images with higher magnification of the upper panel. Scale bar = 100 µm (upper panel) and 50 µm (lower panel). (**C**, **D**) Immunodensities of Iba-1- and GFAP-positive cells. Immunodensities in the graphs were quantified by the Image J program. Data are presented as means ± SEM (two-way ANOVA with Tukey’s post hoc test, F_(7,16)_ = 37.88 (C), F_(7,16)_ = 7.81 (D), *** *p* < 0.001, ** *p* < 0.01, and * *p* < 0.05 versus WT, ns: non-significant).

**Figure 5 jcm-09-00305-f005:**
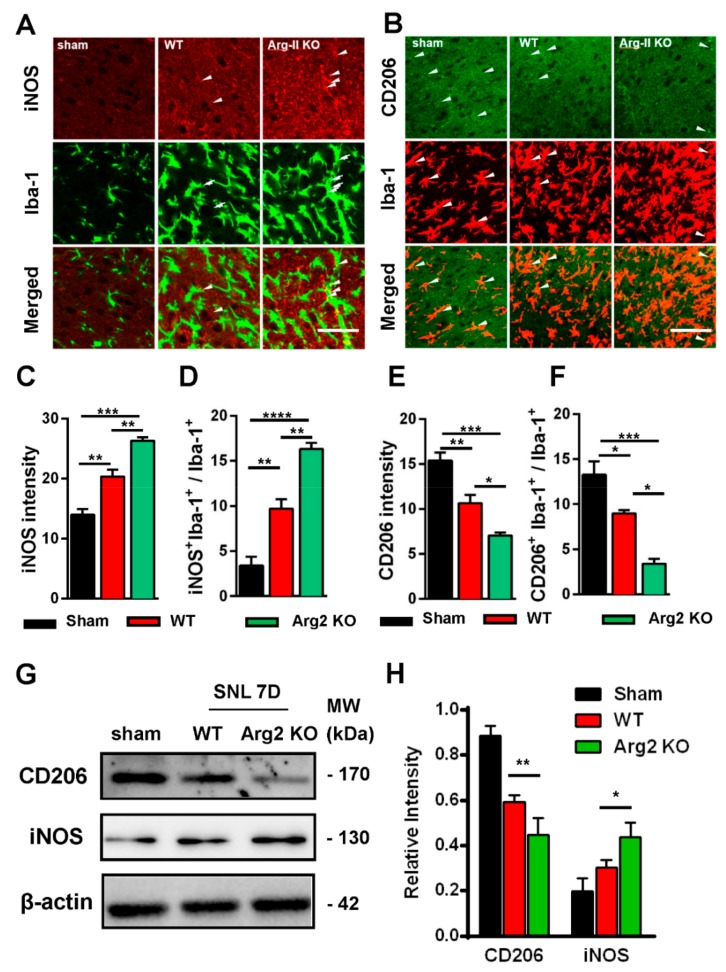
Upregulation of inducible nitric oxide synthase (iNOS) and downregulation of CD206 in the spinal dorsal horn of mice following SNT surgery. (**A, B**) Representative immunohistochemical staining images of spinal cord sections on day 7 post-surgery using iNOS and CD206 antibodies and co-stained with Iba-1 (a microglial marker). The immunoreactivities of iNOS (arrowheads in **A**) and CD206 (arrowheads in **B**) overlapped with Iba-1 microglia. The expression of iNOS increased in WT (*n* = 3) and was notably the highest in Arg2 KO mice (*n* = 3). In contrast, the expression of CD206 decreased, particularly in Arg2 KO mice. Scale bar = 50 µm. (**C, D**) Quantification of iNOS immunodensity and iNOS^+^/Iba-1^+^ co-positive cells/total Iba-1^+^ cells revealed upregulation of iNOS in mice following SNT surgery. (**E, F**) Quantification of CD206 immunodensity and CD206^+^/Iba-1^+^ co-positive cells/total Iba-1^+^ cells exhibited downregulation of CD206 in mice after surgery. (**G, H**) Expression of iNOS and CD206 was assessed by western blotting of the spinal dorsal horns of mice on day 7 post-surgery. Quantification of the blots also indicated upregulation of iNOS and downregulation of CD206. Data are presented as means ± SEM (one-way ANOVA with Tukey’s post hoc test, F_(2,6)_ = 46.15 (**C**), F_(2,8)_ = 47.26 (**D**), F_(2,7)_ = 28.81 (**E**), F_(2,6)_ = 28.86 (**F**), F_(2,2)_ = 0.1073 and F_(1,2)_ = 2.801 (**H**). *** *p* < 0.001, ** *p* < 0.01, and * *p* < 0.05 versus WT or Sham).

**Figure 6 jcm-09-00305-f006:**
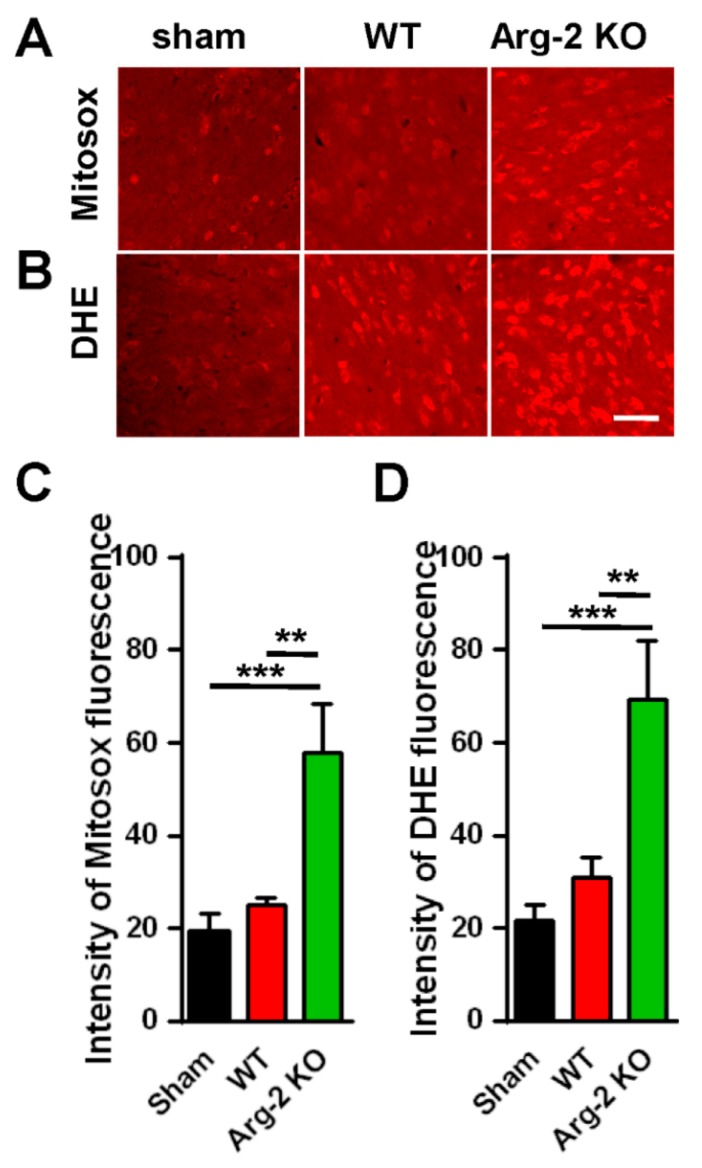
Production of reactive oxygen species (ROS) in the spinal cords of mice following surgery. (**A**, **B**) Upper panel shows the immunohistochemical images of spinal cord sections with the Mitosox ROS antibody, indicating an increased production of ROS in the mitochondria of SNT WT and Arg2 KO mice. The most robust expression of ROS was observed in Arg2 KO mice. The same result was obtained in spinal cord sections stained with Dihydroethidium (DHE), showing the formation of ROS in the cytoplasm (lower panel). Scale bar = 50 µm. (**C**, **D**) ROS and mitochondrial ROS (mtROS) production in the spinal cord of the three groups on day 7 post-surgery was quantified by Image J. Data are presented as means ± SEM (one-way ANOVA with Tukey’s post hoc test, F_(2,6)_ = 31.17 (**C**), F_(2,6)_ = 30.53 (**D**). *** *p* < 0.001 and ** *p* < 0.01 versus Sham).

**Figure 7 jcm-09-00305-f007:**
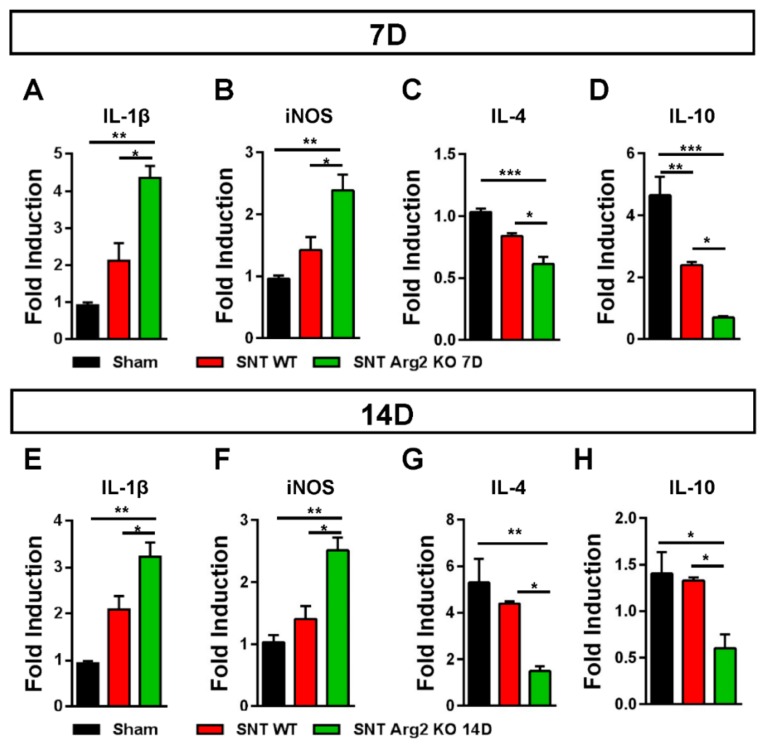
Regulation of inflammatory markers in the mice following the SNT surgery. (**A**, **B**) Gene expression levels of the pro-inflammatory markers interleukin (IL)-1β and iNOS assessed by qRT-PCR indicated an upregulation of either IL-1β or iNOS in the spinal dorsal horns of mice on day 7 after the SNT procedure. (**C**, **D**) In contrast, the levels of the anti-inflammatory markers IL4 and IL-10 on day 7 post-surgery were downregulated in the SNT (*n* = 3 per group) mice. (**E**, **F**, **G**, **H)** Gene expression of these inflammatory markers were also assessed on day 14 post-surgery and obtained similar results. Data are presented as means ± SEM (one-way ANOVA with Tukey’s post hoc test, F_(2,6)_ = 20.75 (**A**), F_(2,6)_ = 14.68 (**B**), F_(2,8)_ = 19.97 (**C**), F_(2,6)_ = 31.68 (**D**), F_(2,7)_ = 19.10 (**E**), F_(2,6)_ = 17.75 (**F**), F_(2,6)_ = 11.14 (**G**), F_(2,6)_ = 7.734 (**H**). *** *p* < 0.001, ** *p* < 0.01, and * *p* < 0.05 versus Sham).

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
