# Peer review of "Arginase 2 Deficiency Promotes Neuroinflammation and Pain Behaviors Following Nerve Injury in Mice"

_jcm, 2020, doi:10.3390/jcm9020305_

Round 1

Reviewer 1 Report

Overall, this study aims to describe the role of Arginase 2 in supporting neuropathic pain states  and microglial activation in the dorsal horn in a mouse model of chronic pain. While the overall study design is reasonable, there are significant redundancies, omissions and a lack of clarity that need substantial revision. In particular, the discussion is hard to follow and there is some discussion of the role of Arg2 in pain related neuroinflammation, but the reader is lost when attempting to compare these findings to prior evidence. I would suggested significantly revising the Intro, Methods, Results and Discussion sections. Specifically, these need revising so that a cogent interpretation can be made with regards to M1 vs M2 microglia activation and the possible role of Asp2 in maintaining pain states in the central nervous system. 

Comments below.  

Minor:

Paper is written in singular first person (i.e. “I confirmed…”), when it should be written in plural (“We confirmed…”) as there are multiple authors.

Methods:

Pain Behavior – foot withdrawal threshold sensing: The authors need to describe their vonFrey filament testing in more details (did they use brushing or punctate delivery of stimuli? What pressure / gauge filament was used?)

There is some evidence that alternative nocifensive behaviors such as paw licking (e.g. number of paw licks per unit time) are more specific for quantifying pain-like behaviors in mice.  The conclusions would be better supported if there was additional converging evidence from other observed pain specific behaviors.  I do appreciate that the Catwalk analysis was used for general locomotor behavioral analysis. Please see Deuis, J. R., Dvorakova, L. S., & Vetter, I. (2017). Methods Used to Evaluate Pain Behaviors in Rodents. Frontiers in molecular neuroscience10, 284. doi:10.3389/fnmol.2017.00284

More details below on vonFrey testing and quantification of immunostaining results.

Additional statistical descriptions needed (detailed below). 

Results and Discussion:

There are multiple occasions where the authors make conclusions / inferences that are not supported by the data. These need to be revised or eliminated from the text. Overall the figures need substantial revising to reduce redundancy, and to make the raw data collected more clear.

For example, on lines 208-210 they state “ The mechanical threshold was considered hyperalgesia  or mechanical allodynia.”  Because the methods of vonFrey testing were not detailed, it is impossible for the reader to know. In general these “pain-like” behaviors are divided into dynamic, punctate or static hyperalgesia or allodynia based on the method used (which was not clearly described).  I presume they are discussed punctate mechanical hyperalgesia.  

-      See Jensen T. S., Finnerup N. B. (2014). Allodynia and hyperalgesia in neuropathic pain: clinical manifestations and mechanisms. Lancet Neurol. 13, 924–935. 10.1016/s1474-4422(14)70102-4

Further, in the next line it reads,” The lower the mechanical threshold observed in the mice, the more sensitive to pain they are.”  This is not entirely accurate. Mechanical thresholds test somatosensory perception, not pain per se. It is likely that in hyperalgesic states, thresholds may be lower to mechanical touch, however the opposite is not true (as currently written).

Lines 230-231 – “ The results obtained from CatWalk analysis were correlated with the von Frey tests.”  This correlation analysis is not shown anywhere. Rephrase.

Lines 240-241 are confusing and difficult to understand what the authors are trying to convey.

Lines 248-249 are an overinterpretation of the data: “Taken together, I concluded that neuropathic pain was successfully induced by the SNT model in both groups of mice.” This is a model of neuropathic pain, and the data do not directly support that neuropathic pain was induced, but rather that behaviors expected from such a model were present.

Lines 396-397  “The levels of inflammatory markers in the spinal cord are direct evidence for the severity of inflammation as well as the pain state.”  This statement is not supported by the evidence. In order to make this claim, you must correlate the pain threshold data with the immunohistochemistry data on the individual mouse level. Specifically, I would recommend a correlation or regression analysis to test that the increased inflammation actually correlates with lower mechanical withdrawal thresholds.

Line 408 is unclear: “This means deficiency of Arg2 imposed effects on both inflammatory reaction and pro-resolution state in the spinal cord.”

Line  436 – Please change or remove the phrase “lower pain threshold” as the authors are not testing pain thresholds, but instead mechanical sensory thresholds as a proxy for hyperalgesia.

Line 440-444, run on sentence please revise.

Line 447 – ‘It is reliable..”   -  The link between Arg 1 downregulation in SNT mice and its relation to Arg2KO is not clear in this sentence.   Please provide a clear description of how Arg2 may be mechanistically driving Arg1 expression.

Line 457 reads “However, I also found data implicating lower production of NO …” Please provide more context to this self-reference to clarify the conflict being discussed.  How does this relate to the influence of Arg2 on iNOS activity?

The discussion in lines 472-482 attempts to reconcile a discrepancy between the classical definition of M1 vs M2 macrophages, but it is very difficult to follow as currently written. Specifically, it is unclear why Arg2 should be considered a marker of M2 activation. Please rewrite this section.

The discussion lacks a description of microanatomy of the dorsal horn, and in which layers or segments microglial changes would be expected.  The most affected nerves for pain transmission are generally in layers II and IV or V,  but immunoexpression patterns (Iga1 and GFAP) appear layer nonspecific. Please address this issue.

Figures:

Fig 1- The data in B appear to be identical to  panel A, with the addition of ARGKO group and different scales on Y axis. There is no need to show Panel A, this information is redundant.  

            - for 2 way ANOVA, it is also customary to report the F statistic with degrees of freedom in the text or caption.

            - Although the authors claim that they conducted a 2 level ANOVA to compare WT vs sham vs Arg2KO groups (one level being group, the other being time, presumably), there is no indication if there was a significant interaction between the two. Fig 1 B seems to suggest that there should be an interaction between group and time effects.

Fig 2 – 2B - The reduction of “Print Area” in WT and Arg2KO could be because reduced ipsilateral paw size, enlarged contralateral paw size, or both. Please show the three data different data types in the same graph for clarity

            2C -  Figure text is not legible. Please improve quality of image.

            2D-  The same criticism applies here as for 2B above. Please show raw data distributions of ‘ipsi and contra’ stance durations.  If panel 2C is actually representative (and mice with Arg2KO almost never used the ipsilateral hind foot, then I am surprised that the mean value for the ratio in figure 2D approaches 25%. These values should be written clearly in the text.  It is also unclear how the data was tabulated for these ratios: is each step cycle considered separately for each mouse and then all cycles tabulated? If not, how is this calculated? The methods section (lines 136-137) does not give sufficient detail.

            The caption lists ‘vs. sham’ at the end, but the comparisons should be between all groups not just sham. This may have been a typo. Also please list the F statistic with degrees of freedom.

Figure 3 – Why do the authors not show Arg2 levels for the Arg2KO mice? This essential validation analysis is missing, however they do provide it for Arg 1 which is less useful.

            3A - Lines 262-263 contains a false statement: “Figure 3A shows that Arg2 mRNA increased the most on day 3 and 7 after surgery and this level was maintained until day 14.” However, the figure directly contradicts this and shows that at day14, the mRNA levels are not significantly different from day 0.

                  3B and 3D – It is not acceptable to show images of western blots alone. The authors should quantify the western blotting group results and analyze them / compare them to the RNA data, instead of showing a single example.

                  The methods simply read “The immunodensities in the graphs were quantified by Image J program software” but the authors need to specify the features and parameters that were used for quantification. Was simple thresholding used (and if so what were the thresholds, how were they decided?) ? If not, was stereological counting used?

Figure 4

                  - It is unclear how much days after surgery immunostaining was done for the sham group. If only 3 days or less,  this  is a possible explanation for low Iba-1 or GFAP in the sham group

                  - What are units for “immunoreactivity”? please label
                   - Can the immunoreactivity be shown for a  region of the dorsal horns that are not expected to change (e.g. Thoracic spine?)              

Figure 5

                  A, B – the example immunofluorographs for iNOS (WT) and CD206 (sham and WT) are not convincing, raising some concern that properly labeled cells are not being counted equally across conditions.  Are there better pictures or are these actually representative?

                  G,H I greatly appreciate that you have quantified the western blots. Please keep the color scheme the same as in C-F.              

Figure 6

                  - similar criticism as for  Figure 3 regarding Image J quantification. Please provide more details in the methods section.

                  - Overall Panels A and B seem to show that Arg2KO has many more ROS than WT or Sham. This seems to contradict the bar graphs showing that Arg2KO has similar fluorescence levels compared to sham for both mitosox and DHE.

Author Response

Overall, this study aims to describe the role of Arginase 2 in supporting neuropathic pain states  and microglial activation in the dorsal horn in a mouse model of chronic pain. While the overall study design is reasonable, there are significant redundancies, omissions and a lack of clarity that need substantial revision. In particular, the discussion is hard to follow and there is some discussion of the role of Arg2 in pain related neuroinflammation, but the reader is lost when attempting to compare these findings to prior evidence. I would suggested significantly revising the Intro, Methods, Results and Discussion sections. Specifically, these need revising so that a cogent interpretation can be made with regards to M1 vs M2 microglia activation and the possible role of Asp2 in maintaining pain states in the central nervous system. 

Comments below.  

Minor:

Paper is written in singular first person (i.e. “I confirmed…”), when it should be written in plural (“We confirmed…”) as there are multiple authors.

Response: we feel sorry for this mistake and thank for your suggestion. We have addressed this in the whole manuscript.

Methods:

Pain Behavior – foot withdrawal threshold sensing: The authors need to describe their vonFrey filament testing in more details (did they use brushing or punctate delivery of stimuli? What pressure / gauge filament was used?)

Response: Thanks for your question. We have added detailed information for how the von Frey filament tests were performed in Materials and Methods part. In detail, the rats were placed on a metal mesh floor covered with clear plastic cages (18 × 8 × 8 cm) and allowed a 20-min period for habituation. Mechanical stimuli were applied with nine different von Frey filaments ranging from 0.008 to 1.4g (0.008, 0.02, 0.04, 0.07, 0.16, 0.4, 0.6, 1 and 1.4 g). Stimuli were applied with a von Frey filament in 3–4-s trials, each of which was repeated four times on each hind paw at approximately 5-min intervals. The 0.6-g filament stimulus was applied first. If a positive response occurred, the next smaller von Frey filament was used; if a negative response occurred, the next larger filament was applied.

There is some evidence that alternative nocifensive behaviors such as paw licking (e.g. number of paw licks per unit time) are more specific for quantifying pain-like behaviors in mice.  The conclusions would be better supported if there was additional converging evidence from other observed pain specific behaviors.  I do appreciate that the Catwalk analysis was used for general locomotor behavioral analysis. Please see Deuis, J. R., Dvorakova, L. S., & Vetter, I. (2017). Methods Used to Evaluate Pain Behaviors in Rodents. Frontiers in molecular neuroscience10, 284. doi:10.3389/fnmol.2017.00284

Response: Thanks for your suggestion. At the present, in addition to the von Frey test and Catwalk gait, our system does not allow us to conduct other analysis for pain behaviors. However, the results obtained from these two analysis are sufficient to evaluate pain behaviors in the mice following the SNT procedure.

 The von Frey test has been widely used to evaluate mechanical stimuli-evoked pain in a neuropathic pain model. The value of 1 g filament is considered the baseline of stimuli threshold and is an innocuous stimulus to the mice under normal condition. That is why, in this test, only mice that pass a baseline von Frey filament test (≥ 1 g) will be selected to undergo the SNT procedure. After the mice are exposed to the SNT surgery, the von Frey tests will be conducted as described above to test the mechanical hypersensitivity induced in the SNT mice. Mechanical hypersensitivity is considered present when mice respond to a ≤ 0.4g filament stimuli. When they only respond to a stimuli ≥1 g, they are considered non sensitive to pain. Importantly, mechanical hypersensitivity (referred to as allodynia) is one of the most common outcomes of neuropathic pain behaviors. As such, the von Frey test has been widely and valuably used in many studies on neuropathic pain models. In addition, we have also performed Catwalk gait for more evidence regarding pain behaviors. The Catwalk gait has been also demonstrated as effective method to evaluate pain behavior in early studies. We have provided reference for this in the manuscript.

Thanks again for your suggestion. According to this, we may attempt to develop other methods to analyze pain behaviors in our future studies.

More details below on vonFrey testing and quantification of immunostaining results.Additional statistical descriptions needed (detailed below). 

 -> Response: We added the detail methods and references

Results and Discussion:

There are multiple occasions where the authors make conclusions / inferences that are not supported by the data. These need to be revised or eliminated from the text. Overall the figures need substantial revising to reduce redundancy, and to make the raw data collected more clear.

For example, on lines 208-210 they state “ The mechanical threshold was considered hyperalgesia  or mechanical allodynia.”  Because the methods of vonFrey testing were not detailed, it is impossible for the reader to know. In general these “pain-like” behaviors are divided into dynamic, punctate or static hyperalgesia or allodynia based on the method used (which was not clearly described).  I presume they are discussed punctate mechanical hyperalgesia.  

-      See Jensen T. S., Finnerup N. B. (2014). Allodynia and hyperalgesia in neuropathic pain: clinical manifestations and mechanisms. Lancet Neurol. 13, 924–935. 10.1016/s1474-4422(14)70102-4n

Further, in the next line it reads,” The lower the mechanical threshold observed in the mice, the more sensitive to pain they are.”  This is not entirely accurate. Mechanical thresholds test somatosensory perception, not pain per se. It is likely that in hyperalgesic states, thresholds may be lower to mechanical touch, however the opposite is not true (as currently written).

Response: we are sorry for these omissions. As described above, we have provided detailed description for methods to evaluate pain behaviors. As responded above, in this study, we used the von frey test to check the mechanical stimuli-evoked pain behavior in the SNT mice. Under normal condition, mice did not respond to an innocuous stimuli (1 g). However, after SNT procedure, they responded to lower values of the filament stimuli. The lowest value obtained from each mouse was recognized as the mechanical threshold of it. These lower thresholds (compared to the baseline ) were only observed in the mice after they underwent the SNT procedure, suggesting that the SNT procedure successfully induced mechanical hypersensitivity in these mice. And the responses of the mice to various filament stimuli were different from each other. This indicated the different mechanical hypersensitivity state among these mice. In order to avoid concerns from the readers, we have also carefully revised the text when referring to the results obtained in the behavior test.

Lines 230-231 – “ The results obtained from CatWalk analysis were correlated with the von Frey tests.”  This correlation analysis is not shown anywhere. Rephrase.

Response: This sentence means the results obtained from the Catwalk analysis also indicate that the pain state observed in the Arg2 KO mice was more serious than this in the WT mice. And this finding was consistent with the results in the von Frey test, showing lower mechanical thresholds of the SNT Arg2 KO mice.

We have revised the text to avoid confusion from the readers.

Lines 240-241 are confusing and difficult to understand what the authors are trying to convey.

Response: we are sorry. We have rephrased the text more clearly.

Lines 248-249 are an overinterpretation of the data: “Taken together, I concluded that neuropathic pain was successfully induced by the SNT model in both groups of mice.” This is a model of neuropathic pain, and the data do not directly support that neuropathic pain was induced, but rather that behaviors expected from such a model were present.

Response: Thanks for showing us this confusion. We have revised it in the manuscript.

Lines 396-397  “The levels of inflammatory markers in the spinal cord are direct evidence for the severity of inflammation as well as the pain state.”  This statement is not supported by the evidence. In order to make this claim, you must correlate the pain threshold data with the immunohistochemistry data on the individual mouse level. Specifically, I would recommend a correlation or regression analysis to test that the increased inflammation actually correlates with lower mechanical withdrawal thresholds.

Response: Our data indeed support the literature. In the von Frey test, we observed the lower mechanical thresholds in the SNT Arg2 KO mice, compared to the WT. Since it has been demonstrated that neuropathic pain usually results from chronic neuroinflammation and the pain state is often consistent with inflammatory condition after nerve injury, it is reasonable to further investigate the inflammatory state following confirmation of pain behaviors. So that, to address the question whether the more severe pain state observed in the KO mice was related to the inflammatory condition in the spinal cord after nerve injury, we attempted to assess and compare inflammatory condition between the WT and the KO mice. The inflammatory condition was indicated by the levels of microglial activation and inflammatory mediators released in the spinal cord. For the roles of microglia and their contribution to the development of neuroinflammation, we have presented and provided references in the manuscript. As expected, the stronger activation of glial cells and higher levels of proinflammatory mediators were found in the SNT KO mice, suggesting the more excessive neuroinflammation in the KO mice after the SNT procedure. Regarding your recommendation that we should correlate the pain threshold data with the immunohistochemistry data on the individual mouse level, we can explain that we aimed to compare the condition between the ArgKO and the WT mice. The lower mechanical thresholds of the KO mice correlate with the results in staining data and the higher levels of proinflammatory markers, as described above. There was significant difference between the SNT Arg2 KO and the SNT WT mice when compared. But there was no significant difference among the mice in one group. We collected the mice that presented the quite similar mechanical thresholds in each group for staining or RT PCR analysis. As such, the results mainly indicate the difference between the KO and WT mice and no difference among the mice in one group.

Line 408 is unclear: “This means deficiency of Arg2 imposed effects on both inflammatory reaction and pro-resolution state in the spinal cord.”

Response: because the levels of proinflammatory mediators are regarded as the development of inflammatory reaction. Meanwhile, the presence of antiinflammatory mediators is a sign of the acquisition of the healing or resolution process. When compared to the WT mice, the SNT KO mice showed both higher levels of proinflammatory mediatorss and lower levels of antiiflammatory mediatorss. As such, that statement means the lack of Arg2 in the KO mice affected both the levels of pro and anti-inflammatory mediators.

Line  436 – Please change or remove the phrase “lower pain threshold” as the authors are not testing pain thresholds, but instead mechanical sensory thresholds as a proxy for hyperalgesia.

Response: thanks for your reminder. We have revised it in the manuscript according to your suggestion.

Line 440-444, run on sentence please revise.

Response: thanks for your reminder. We have addressed this.

Line 447 – ‘It is reliable..”   -  The link between Arg 1 downregulation in SNT mice and its relation to Arg2KO is not clear in this sentence.   Please provide a clear description of how Arg2 may be mechanistically driving Arg1 expression.

Response: Arg1 and Arg2 are two isoforms of Arginase enzyme system and share the similar function in l-arginine metabolism. Arg1 has been demonstrated in numerous early studies to promote M2 macrophage marker but Arg2 remains unclear. We conducted experiment to verify the basal levels of Arg1 in both of the sham WT and the sham Arg2 KO and regulation of Arg1 in the injury groups following SNT procedure. As indicated in the manuscript, the basal level of Arg1 in the sham of the Arg2 KO remained unchanged compared to the sham WT, indicating that the deletion of Arg2 in the KO mice did not affect the level of Arg1 under normal condition. However, after the SNT procedure, Arg1 was downregulated in the spinal dorsal horn of the injury mice, compared to the sham. Of note, Arg1 was higher downregulated in the SNT Arg2 KO than in the SNT WT, compared with the corresponding sham groups (the sham WT and the sham Arg2 KO). This indicates KO of Arg2 directly affected the abundance mRNA of Arg1 following surgery. Alternatively, Arg2 defficiency resulted in higher reduction of Arg1 induced by inflammation in the spinal cord, suggesting a modulatory role for Arg2. Our study has suggested a role of Arg2 for the first time in a neuropathic pain model.

Line 457 reads “However, I also found data implicating lower production of NO …” Please provide more context to this self-reference to clarify the conflict being discussed.  How does this relate to the influence of Arg2 on iNOS activity?

Response: In that study, they have demonstrated that Arg2 overexpression in mouse joint tissues causes OA with synovitis and loss of glycosaminoglycans in articular cartilage followed by upregulation of pro-inflammatory products, although, their data show ablation of Arg2 promoted IL-1b – caused NO production. However, this enhancement of NO production in Arg2−/−chondrocytes is not associated with the ability of Arg2 to regulate OA pathogenesis. As such, in the pathogenesis of OA, they have suggested a detrimental role for Arg2. In contrast to our concept, evidence that a lack of Arg2 could promote NO production may suggest a beneficial role of Arg2 in the pathogenesis of neuropathic pain. We have mentioned this in our manuscript.

The discussion in lines 472-482 attempts to reconcile a discrepancy between the classical definition of M1 vs M2 macrophages, but it is very difficult to follow as currently written. Specifically, it is unclear why Arg2 should be considered a marker of M2 activation. Please rewrite this section.

Response: Thanks for your comment. In fact, many previous studies have provided the data using knockdown or knockout of gene encoding for the marker of interest to investigate the possible roles of them. And probably, when a lack of one marker leads to more serious pathological condition, along with evidence that overexpression of it contributes to resolution, this marker would be recognized to play beneficial roles. Our data demonstrate that a lack of Arg2 resulted in more severe inflammatory state in the spinal cord and lower mechanical thresholds in the SNT Arg2 KO mice, compared to the SNT WT. This directly suggests Arg2 deficiency contributes to inflammation and also indicates that Arg2 plays beneficial role in neuroinflammation.

We have revised the text according to your suggestion.

The discussion lacks a description of microanatomy of the dorsal horn, and in which layers or segments microglial changes would be expected.  The most affected nerves for pain transmission are generally in layers II and IV or V,  but immunoexpression patterns (Iga1 and GFAP) appear layer nonspecific. Please address this issue.

Response: in the pathogenesis of neuropathic pain, particularly in the L5 SNT-induced neuropathic pain model, microglia or astrocytes are usually activated the most in laminae 1 and 2 of the spinal dorsal horn, segments L4-L5 following nerve injury. Thanks for your question and we have addressed this in our revised manuscript.

Figures:

Fig 1- The data in B appear to be identical to  panel A, with the addition of ARGKO group and different scales on Y axis. There is no need to show Panel A, this information is redundant.  

Response: We agree with you that we don’t need to show both of figure 1A and 1B. However, when the two figures are displayed together, it seems that the more severe pain induced in the Arg2 KO mice is more clearly implicated. According to your suggestion, we have revised Figure 1: fig1 A, B now show a uniform axis so the mechanical thresholds of each group are clearly indicated

            - for 2 way ANOVA, it is also customary to report the F statistic with degrees of freedom in the text or caption.

            - Although the authors claim that they conducted a 2 level ANOVA to compare WT vs sham vs Arg2KO groups (one level being group, the other being time, presumably), there is no indication if there was a significant interaction between the two. Fig 1 B seems to suggest that there should be an interaction between group and time effects.

Response: thanks for your suggestion. We have complemented the F values with degrees of freedom in the figure legends.

Fig 2 – 2B - The reduction of “Print Area” in WT and Arg2KO could be because reduced ipsilateral paw size, enlarged contralateral paw size, or both. Please show the three data different data types in the same graph for clarity

Response: Thanks for your comment. The data we provided in fig 2B is the representative one. Since the size of the images is too big to arrange in one figure so we would like to show only the representative one. We have added two other images of Print Area for additional data in supplementary figure 1, as you suggested. Please, refer to the supplementary figure.. And please be aware that we did not make any changes to the original images that were obtained from the CatWalk program software.

            2C -  Figure text is not legible. Please improve quality of image.

Response: Thanks for your comment. We have addressed this in the revised manuscript.

            2D-  The same criticism applies here as for 2B above. Please show raw data distributions of ‘ipsi and contra’ stance durations.  If panel 2C is actually representative (and mice with Arg2KO almost never used the ipsilateral hind foot, then I am surprised that the mean value for the ratio in figure 2D approaches 25%. These values should be written clearly in the text.  It is also unclear how the data was tabulated for these ratios: is each step cycle considered separately for each mouse and then all cycles tabulated? If not, how is this calculated? The methods section (lines 136-137) does not give sufficient detail.

Response: thanks for your comment. The quantitative data for print area or single stance for each mouse was automatically calculated by the CatWalk program software and then was tabulated to create the graph based on three or four independent data. The images shown in the figure are the representative ones. We have provided two other images for analysis of Single Stance in supplementary figure 1. We have also indicated the values of these proportions in the text.

            The caption lists ‘vs. sham’ at the end, but the comparisons should be between all groups not just sham. This may have been a typo. Also please list the F statistic with degrees of freedom.???

Response: thanks for your kind comment. We have addressed this in the figure legends.

Figure 3 – Why do the authors not show Arg2 levels for the Arg2KO mice? This essential validation analysis is missing, however they do provide it for Arg 1 which is less useful.

Response: Thanks for your suggestion. According to your comment, we have complemented the data for protein expression of Arg2 in the KO mice (additional data shown in Fig 3B) and this data serves as confirmation of KO of Arg2. As we discussed in our manuscript, the assessment of Arg1 expression is essential to verify which one of these two isoforms directly contributed to the results that we observed.

           3A - Lines 262-263 contains a false statement: “Figure 3A shows that Arg2 mRNA increased the most on day 3 and 7 after surgery and this level was maintained until day 14.” However, the figure directly contradicts this and shows that at day14, the mRNA levels are not significantly different from day 0.

Response: we feel sorry for this mistake. We have rephrased this sentence.

                  3B and 3D – It is not acceptable to show images of western blots alone. The authors should quantify the western blotting group results and analyze them / compare them to the RNA data, instead of showing a single example.

Response: thanks for your comment. We have added quantification of band intensity for western blots in figures 3B, 3D.

                  The methods simply read “The immunodensities in the graphs were quantified by Image J program software” but the authors need to specify the features and parameters that were used for quantification. Was simple thresholding used (and if so what were the thresholds, how were they decided?) ? If not, was stereological counting used?

Response: when using the Image J Program software, the same size of selected area (usually laminae I and II) in each slice was cropped, then the immunodensity was caculated following the ideal setting. The data are presented as mean value of the immunoreactivity. The graphs were created based on the data analyzed from three independent slices.

Figure 4

                  - It is unclear how much days after surgery immunostaining was done for the sham group. If only 3 days or less,  this  is a possible explanation for low Iba-1 or GFAP in the sham group

                  - What are units for “immunoreactivity”? please label 
                   - Can the immunoreactivity be shown for a  region of the dorsal horns that are not expected to change (e.g. Thoracic spine?)

Response: in figure 4, the spinal cord sections of the sham were collected on day 3 post injury. In our experimental model, the sham group, in which mice underwent the similar surgical procedure to the SNT surgery accepting L5 spinal nerve transection, are referred to as the naïve group. Because the mice in the sham group usually do not show any significant difference in mechanical hypersensitivity, compared to the naïve mice. Accordingly, we can not observe any changes in glial population in the spinal cord of the sham mice following surgery. The similar low immunoreactivities of iba-1 and GFAP are often recorded when we compare those between the ipsilateral and contralateral sides of the spinal dorsal horn of the sham. And there is no significant difference when we check it at different time points 3, 7, or 14 days post surgery. This is because the sham surgical procedure does not cause nerve injury and does not evoke glial activation at all.

As responded above, the lumbar enlarged (L4-L6) regions of the spinal cord were removed and processed for immunostaining. Then the stained slices were imaged at the laminae I+II region where we usually observe the most robust changes of glial cells.              

Figure 5

                  A, B – the example immunofluorographs for iNOS (WT) and CD206 (sham and WT) are not convincing, raising some concern that properly labeled cells are not being counted equally across conditions.  Are there better pictures or are these actually representative?

Response: These are the best staining of iNOS and CD206. Because these markers are very difficult to perform staining with in vivo slices. In the garphs of the total immunodensities of iNOS and CD206, the immunodensity was quantified by Image J Program software as described above. The graphs of iNOS+ or CD206+ + iba-1+/ iba-1+ display the proportions (%) of these corresponding expressing cells. As presented in the graphs, these proportions (not the number of cells) are very low. The data were analyzed from three independent slices.   

                  G,H I greatly appreciate that you have quantified the western blots. Please keep the color scheme the same as in C-F.

Response: thanks for your kind comment. We have revised the color scheme in figure 5H.              

Figure 6

                  - similar criticism as for  Figure 3 regarding Image J quantification. Please provide more details in the methods section.

Response: Thanks and please refer to the response above.

                  - Overall Panels A and B seem to show that Arg2KO has many more ROS than WT or Sham. This seems to contradict the bar graphs showing that Arg2KO has similar fluorescence levels compared to sham for both mitosox and DHE.

Response: we are so sorry for this careless mistake. We got confused between the WT bar and the Arg2 KO bar. Thanks so much for showing us this confusion. We have revised the graphs accordingly.

Reviewer 2 Report

Dear Editor,

The manuscript ‘Arginase 2 deficiency promotes neuroinflammation and pain behaviors following nerve injury in mice’ by Dr Yin and coworkers investigates the functional role of the enzyme arginase 2 (Arg2) in neuroinflammation and associated pain in a murine model of experimental nerve injury (spinal nerve transection, SNT).

For the purpose, the authors utilise an Arg2 global KO mice. Following SNT or sham surgery in WT and Arg2 KO, the authors conduct behavioural experiments over a period of 14 days to establish the degree of mechanical allodynia (Von Frey filaments) following the induction of experimental neuropathy. They also assess for the appearance of gait alterations (Catwalk analysis). From a neurochemical perspective, initial experiments aim at establishing whether Arg2 is upregulated in the dorsal horns by SNT in WT mice and then test if the cognate enzyme Arg1 is deregulated between the two genotypes. Further experiments include evaluating the differential level of astrogliosis, microgliosis and ROS production in the spinal cord of WT Vs Arg2 KOs. Finally, the portray how certain pro-inflammatory (IL1beta, iNOS) and anti-inflammatory factors (IL4 and IL10) are regulated in the spinal cords of Arg2 KO mice at the time points tested (0, 7 and 14 days post-SNT).

The findings show that following SNT, Arg2 KO mice display significantly higher mechanical hypersensitivity than WTs. These results are corroborated by reductions in the expression of Arg1 (a known anti-inflammatory factor) in the dorsal horns and heightened astrocytosis, microgliosis, ROS accumulation and pro-inflammatory cytokines in the spinal cord of Arg2 deficient mice. Conversely, anti-inflammatory cytokines are more robustly reduced in Arg2 KO mice than in WTs.

This is an interesting study addressing the contribution of Arg2 to the development of neuropathy and behavioural and neurochemical sequelae. Nonetheless, this reviewer identified a series of flaws that are not convincing and warrant further investigations.

(Figure 1A) I would recommend to utilise the same y-axis settings as in Panel B for better visual comparison. (Figure 2) It is not clear why gait analyses were performed only at 7 days post-SNT whereas all the experiments were thoroughly conducted at 7 and 14 days post-injury. (Figure 3B and 3D) How were Western blot analyses carried out to obtain n=3 per group? This reviewer encourages repeating the experiments with more animals per group (n=4-6) and uploading representative images with more than a protein sample per group. Additionally, please add MW next to each blot. (Figure 4A and 4B) From the pictures shown it is hard to decipher the difference between Iba1 and GFAP staining. Also, scale bars are missing from both the main image and insets. Higher resolution images are recommended. (Figure 5G) Please add MW next to each blot. (Figure 6A and 6B) Scale bars missing. This reviewer discourages the use of the ‘first person’ in a manuscript as this is (usually) the combined effort of multiple investigators. Given the number of co-authors, this paper is not an exception. (Microdissection of the ipsilateral dorsal horn) Obtaining protein lysates from such a discreet region is a daunting task. The authors should provide more details on how they managed to remove the fresh spinal cord and dissect out the dorsal horn corresponding to the spinal segment that is the recipient of the transected nerve.

Author Response

The manuscript ‘Arginase 2 deficiency promotes neuroinflammation and pain behaviors following nerve injury in mice’ by Dr Yin and coworkers investigates the functional role of the enzyme arginase 2 (Arg2) in neuroinflammation and associated pain in a murine model of experimental nerve injury (spinal nerve transection, SNT).

For the purpose, the authors utilise an Arg2 global KO mice. Following SNT or sham surgery in WT and Arg2 KO, the authors conduct behavioural experiments over a period of 14 days to establish the degree of mechanical allodynia (Von Frey filaments) following the induction of experimental neuropathy. They also assess for the appearance of gait alterations (Catwalk analysis). From a neurochemical perspective, initial experiments aim at establishing whether Arg2 is upregulated in the dorsal horns by SNT in WT mice and then test if the cognate enzyme Arg1 is deregulated between the two genotypes. Further experiments include evaluating the differential level of astrogliosis, microgliosis and ROS production in the spinal cord of WT Vs Arg2 KOs. Finally, the portray how certain pro-inflammatory (IL1beta, iNOS) and anti-inflammatory factors (IL4 and IL10) are regulated in the spinal cords of Arg2 KO mice at the time points tested (0, 7 and 14 days post-SNT).

The findings show that following SNT, Arg2 KO mice display significantly higher mechanical hypersensitivity than WTs. These results are corroborated by reductions in the expression of Arg1 (a known anti-inflammatory factor) in the dorsal horns and heightened astrocytosis, microgliosis, ROS accumulation and pro-inflammatory cytokines in the spinal cord of Arg2 deficient mice. Conversely, anti-inflammatory cytokines are more robustly reduced in Arg2 KO mice than in WTs.

This is an interesting study addressing the contribution of Arg2 to the development of neuropathy and behavioural and neurochemical sequelae. Nonetheless, this reviewer identified a series of flaws that are not convincing and warrant further investigations.

Response: thank you so much for your kind comment. We appreciate it very much.

(Figure 1A) I would recommend to utilise the same y-axis settings as in Panel B for better visual comparison.

Response: thanks for your suggestion. We have addressed this in the revised manuscript.

 (Figure 2) It is not clear why gait analyses were performed only at 7 days post-SNT whereas all the experiments were thoroughly conducted at 7 and 14 days post-injury.

Response: actually, in the SNT-induced neuropathic pain model, the SNT-induced mechanical hypersensitivity, which is measured and assessed by the von Frey test, is the most common symptom. And this has been used to evaluate and monitor the duration of pain behavior following nerve injury in many neuropathic pain models. Besides, the CatWalk gait has been used to assess another sign of pain: the motor function that is affected by spontaneous pain after nerve injury. In this study, we assessed SNT-induced neuropathic pain behaviors mainly based on the von Frey test, which was monitored at various time points up to day 14 post injury. The CatWalk gait was also assessed in addition to the von Frey test to support pain behavior test. However, this test was performed only on day 7 post injury, when we usually observe the greatest changes of glial cells in the spinal cord along with serious pain behaviors.

 (Figure 3B and 3D) How were Western blot analyses carried out to obtain n=3 per group? This reviewer encourages repeating the experiments with more animals per group (n=4-6) and uploading representative images with more than a protein sample per group. Additionally, please add MW next to each blot.

Response: the western bots shown in the figures are the representative ones. We have repeated each experiment at least three times. Thanks for your suggestion and we have increased the number of animal in one group up to 4 and complemented the quantification for band intensity in figure 3B, D.

 (Figure 4A and 4B) From the pictures shown it is hard to decipher the difference between Iba1 and GFAP staining. Also, scale bars are missing from both the main image and insets. Higher resolution images are recommended.

Response: indeed, when using iba1 (microglial marker) and GFAP (astrocytic marker) to recognize microglia and astrocytes in immunostained sections, the morphology of microglia and astrocytes will be visualized. These glial cells share many similar morphological characteristics. As such, the positive immunoraectivities of these markers look quite similar. But they are independent of each other. We have also added scare bars and heightened the resolution of the images.

(Figure 5G) Please add MW (molecular weight) next to each blot. (Figure 6A and 6B) Scale bars missing.

Response: thanks for your suggestion. We have addressed all in the figures.

This reviewer discourages the use of the ‘first person’ in a manuscript as this is (usually) the combined effort of multiple investigators. Given the number of co-authors, this paper is not an exception.

Response: thanks for your suggestion. We have addressed this in the revised manuscript.

(Microdissection of the ipsilateral dorsal horn) Obtaining protein lysates from such a discreet region is a daunting task. The authors should provide more details on how they managed to remove the fresh spinal cord and dissect out the dorsal horn corresponding to the spinal segment that is the recipient of the transected nerve. 

Response: Thanks for your suggestion. We have provided detailed information for how the spinal dorsal horns were collected from the mice, as follows:

Mice were sacrificed on day 7 after the SNT surgery with sodium pentobarbital (50 mg/kg, i.p) Shortly afterwards, the vertebral column was exposed and was then removed from the mice. The spinal cord tissue was extracted using hydraulic pressure applied to the caudal vertebral canal, whereupon the tissue was washed in PBS to remove remained blood contaminants. The enlarged segments L4-L5 (0.5 cm)  were carefully dissected out. The halves of lumbar segments ipsilateral to the SNT or sham surgery were collected. The dorsal part of the spinal cord was separated from the ventral part. Afterwards, the tissues were quickly frozen in nitrogen liquid and then stored at −20 °C until analyses.

Round 2

Reviewer 1 Report

Your responses to my comments were nearly all addressed, and I greatly appreciate your quick turnaround. Overall, the methods are more clearly described, the text and figures are more lucid, and the overall presentation is more comprehensive as to what was accomplished. This is a novel addition to the field of neuroinflammation and pain and almost there.

There are some still some minor concerns: 

1.  Fig 1A is unnecessary, please remove it.  (it would work fine in a presentation, but there is  no need in a published paper to print  the data twice)

2. There are multiple sentences which are missing prepositions or the singular is used when plural is needed. A good copy editing is needed for the whole manuscript, to ensure that minor points are not missed due to improper English use. This may be corrected in proof editing. 

Some examples (but not all include) 

Lines:

23 - "a beneficial or detrimental role"

302-305 "Higher downregulation" sounds murky - would be clearer as  "stronger downregulation" or "increased downregulation"

433- "my assessment" should be 'our assessment'

438 - "7 day tissues"  should read  "day 7."

475 - 'experiment' should be plural 'experiments'

Author Response

Dear Editors and Reviewers,

We appreciate the reviewers’ comments and suggestions very much.

 We have followed the instructions and have addressed most of the concerns in our revised manuscript.  As your recommendation, we have deleted Fig 1A and changed the manuscript accordingly.

Regarding the concern of English editing, we have made some corrections to the manuscript as the reviewer suggested. In addition, please be informed that our manuscript has undergone extensive English editing provided by professional service company (Textcheck) before we submitted it. For certificate, please refer to the link below:

http://www.textcheck.com/certificate/n9TzVG

Once again, we would like to express sincere thanks to the reviewers for your valuable suggestions that make our manuscript more comprehensive. We hope that this revised manuscript will meet the requirements for publication of this journal. Thank you!

Reviewer 2 Report

Dear Editor,

The authors have addressed and/or provided reasonable explanations for the issues raised by this reviewer. The manuscript has substantially improved in quality. No further comments or criticism.

Author Response

(The authors gave the same response as above.)
